# Enhancing $CO_2$ electrolysis through synergistic control of non-stoichiometry and doping to tune cathode surface structures

Lingting Ye[1,*], Minyi Zhang[2,*], Ping Huang[1], Guocong Guo[2], Maochun Hong[2], Chunsen Li[2], John T.S. Irvine[1,3] & Kui Xie[1]

Sustainable future energy scenarios require significant efficiency improvements in both electricity generation and storage. High-temperature solid oxide cells, and in particular carbon dioxide electrolysers, afford chemical storage of available electricity that can both stabilize and extend the utilization of renewables. Here we present a double doping strategy to facilitate $CO_2$ reduction at perovskite titanate cathode surfaces, promoting adsorption/activation by making use of redox active dopants such as Mn linked to oxygen vacancies and dopants such as Ni that afford metal nanoparticle exsolution. Combined experimental characterization and first-principle calculations reveal that the adsorbed and activated $CO_2$ adopts an intermediate chemical state between a carbon dioxide molecule and a carbonate ion. The dual doping strategy provides optimal performance with no degradation being observed after 100 h of high-temperature operation and 10 redox cycles, suggesting a reliable cathode material for $CO_2$ electrolysis.

[1] Key Lab of Design & Assembly of Functional Nanostructure, Fujian Institute of Research on the Structure of Matter, Chinese Academy of Sciences, Fuzhou, Fujian 350002, China. [2] State Key Lab of Structural Chemistry, Fujian Institute of Research on the Structure of Matter, Chinese Academy of Sciences, Fuzhou, Fujian 350002, China. [3] School of Chemistry, University of St Andrews, St Andrews, Fife KY16 9ST, Scotland, UK. * These authors contributed equally to this work. Correspondence and requests for materials should be addressed to C.L. (email: chunsen.li@fjirsm.ac.cn) or to J.T.S.I. (email: jtsi@st-andrews.ac.uk) or to K.X. (email: kxie@fjirsm.ac.cn).

Solid oxide electrolysers (SOEs) have been attracting great interest due to their high efficiencies in converting low-carbon fuels from renewable electrical energy[1,2]. They can exploit available high-temperature heat streams from nuclear plants or exhaust industry heat to maximize electrical efficiency and both thermodynamic and kinetic advantages can be anticipated because of the high operating temperatures[3,4]. In SOEs, using an externally applied potential, $CO_2$ can be electrochemically converted into CO and $O^{2-}$ at the cathode, while the generated $O^{2-}$ ions transport through the electrolyte to the anode to form $O_2$ gas[1,5,6].

Currently, nickel/yttria-stabilized zirconia (Ni-YSZ) composites are the cathode of choice for high-temperature SOEs[5,6]. In such a composite, the percolating networks of both Ni and YSZ provide sufficient electronic and ionic conductivity, while the Ni guarantees high electrocatalytic activity towards the reduction reaction. Long-term operation with Ni-YSZ is feasible only in $CO/CO_2$ gas mixtures, where the presence of CO maintains a reducing atmosphere[7]. Under realistic conditions however, reduction–oxidation (redox) cycles of Ni will inevitably occur in the cathode, ultimately leading to electrode degradation and delamination[8,9]. In contrast, redox-stable ceramic cathodes would offer a promising alternative for direct high-performance $CO_2$ electrolysis. Especially materials exhibiting n-type conduction properties are expected to demonstrate improved conductivity under the strongly reducing cathode conditions. Perovskite-type doped strontium titanates, $(La,Sr)TiO_{3+\delta}$ ($LST_{O+}$), are such materials, due to the reducibility of $Ti^{4+}$ to $Ti^{3+}$, and have therefore attracted a significant amount of interest within the field of SOE and fuel cell electrodes[10,11]. A composite cathode based on $La_{0.2}Sr_{0.8}TiO_{3.1}$ was shown to be well adapted to direct $CO_2$ electrolysis[12], because the titanate is partially electrochemically reduced ($Ti^{4+} \rightarrow Ti^{3+}$) at potentials required for $CO_2$ reduction and the n-type electronic conduction is accordingly enhanced, but cathode performance for $CO_2$ electrolysis is still limited by insufficient electro-catalytic activity and the weak high-temperature chemical adsorption of reactants[13].

The incorporation of catalytically active metal nanoparticles through impregnation methods has proven to be an effective approach to enhance ceramic electrode activity[14]. However, long-term stability of nanocatalysts at high operating temperature remains a major challenge[15] due to particle agglomeration leading to performance degradation[16,17]. An alternative method is to incorporate the metal element as a dopant within the host lattice during the synthesis of the catalyst in air, which is then exsolved at the surface in the form of catalytically active metallic nanoparticles under reducing conditions. If the composition and conditions are carefully chosen to avoid full decomposition, anchored nanoparticles can be grown on the cathode. Any possible agglomeration of exsolved Ni nanoparticles on the substrate surface can be remedied by periodically cycling from oxidizing to reducing conditions[11]. We have recently demonstrated the *in situ* growth of metal nanoparticles directly from a perovskite backbone support through control of composition, particularly by tuning deviations from the ideal $ABO_3$ stoichiometry[18]. The exsolved metal nanoparticles exhibit enhanced high-temperature stability and improved coking resistance, due to a stronger metal–oxide interface resulting from an anchoring effect with the parent perovkiste. The key surface effects and defect interactions of exsolution-based perovskite materials are expected to demonstrate promising catalytic functionalities[19].

High-temperature $CO_2$ electrolysis suffers from poor adsorption and activation of the reactant, due to the linear molecules lacking polarity. This is believed to cause local starvation of $CO_2$ in SOE cathodes[1,3,7,12]. Currently, preferential chemical adsorption of $CO_2$ on solid oxide materials is based on grafting solid amines, which produces an alkaline surface. However, the desorption temperature is normally below 500 °C, which is far below typical electrolyser operating temperatures[20,21]. Surface oxygen vacancies created next to redox active sites on solid oxide materials may provide alternative sites for $CO_2$ chemisorption and are expected to significantly elevate the onset temperature of chemical desorption of $CO_2$, which would benefit cathode performance[22]. Combining oxygen vacancies and exsolved catalytic nanoparticles should finally produce an active interface for electrocatalytic $CO_2$ reduction.

In this work, active nanostructures are investigated on titanate surface under differing regimes of perovskite non-stoichiometry. The exsolved metal nanoparticles coupled with tailored oxygen vacancies through Cr and Mn substitution produce a strongly interactive interface. Chemical adsorption/activation of $CO_2$ is investigated on these titanate surfaces as well as their effectiveness as cathode materials for indirect $CO_2$ electrolysis.

## Results

**Crystal structure and microstructure.** Different perovskite-defect chemistries and their combinations have been investigated, (oxygen excess, cation deficient and Cr/Mn doped) seeking to optimize electrolysis cathode properties. X-ray diffraction analysis confirms the synthesis of pure-phase $La_{0.2}Sr_{0.8}Ti_{1.0}O_{3+\delta}$ ($LST_{O+}$), $La_{0.2}Sr_{0.8}Ti_{0.9}Cr_{0.1}O_{3+\delta}$ ($LSTC_{O+}$), $La_{0.2}Sr_{0.8}Ti_{0.9}Mn_{0.1}O_{3+\delta}$ ($LSTM_{O+}$), $(La_{0.2}Sr_{0.8})_{0.95}Ti_{0.9}Mn_{0.1}O_{3+\delta}$ ($LSTM_{A-}$), $(La_{0.2}Sr_{0.8})_{0.95}Ti_{0.9}Cr_{0.1}O_{3-\delta}$ ($LSTC_{A-}$), $(La_{0.2}Sr_{0.8})_{0.95}Ti_{0.85}Cr_{0.1}Ni_{0.05}O_{3+\delta}$ ($LSTCN_{A-}$) and $(La_{0.2}Sr_{0.8})_{0.95}Ti_{0.85}Mn_{0.1}Ni_{0.05}O_{3+\delta}$ ($LSTMN_A$) powders (Fig. 1). The patterns are shown in Supplementary Section (Supplementary Fig. 1a,b). All oxidized samples could be indexed assuming a cubic symmetry, with space group $Pm$-$3m$. As shown in Supplementary Figs 1 and 2 and Supplementary Table 1, on reduction, a unit cell expansion is observed due to the reductions of Ti, Mn and Cr (from $M^{4+}$ to $M^{3+/2+}$). The doping and reduction of Mn/Cr should be coupled with the creation of additional oxygen vacancies in the titanate, which in turn should facilitate $CO_2$ chemisorption. The $LST_{O+}$ is oxygen-excess while the oxygen with $\delta$ amount is present in the form of oxygen interstitial in perovskite oxide. However, the $(La,Sr)(Ti,M)O_{3+\delta}$ (M = Mn, Cr) has redox-active dopants such as Mn linked to oxygen vacancies even though the interstitial oxygen might be still present after reduction. The nickel containing compounds $LSTMN_{A-}$ and $LSTCN_{A-}$ exhibit nickel exsolution on reduction, as observed in X-ray diffraction, X-ray photoelectron spectroscopy (XPS) and microscopy. Supplementary Fig. 2 shows the emergence of a peak at $2\theta = 44.5°$, corresponding to cubic metallic nickel. Due to the A-site deficiency, the perovskite structure is retained for the titanate phase, suggesting structural redox stability. XPS on oxidized and reduced samples shows that nickel is only present as $Ni^{2+}$ in the former, whereas metallic $Ni^0$ is additionally present in the latter, corroborating the X-ray diffraction results. XPS furthermore confirms the presence of the reduced species $Ti^{3+}$, $Mn^{3+}$ and $Cr^{3+/2+}$ within the reduced titanates (Supplementary Figs 3 and 4). Figure 2a,b present scanning electron microscopic micrographs of sintered $LSTMN_{A-}$ pellets after reduction, showing uniform titanate surface decoration with Ni nanoparticles. The nanoparticles exist within a narrow size distribution, with an average of 60 nm, which should prove highly active as electrocatalyst. The growth of these Ni nanoparticles was studied using high-resolution transmission electron microscopy. From the lattice spacings in Fig. 2c,d, 0.275 and 0.203 nm, it becomes clear that two distinct phases have emerged, that is, reduced titanate (110) and metallic nickel (111), respectively.

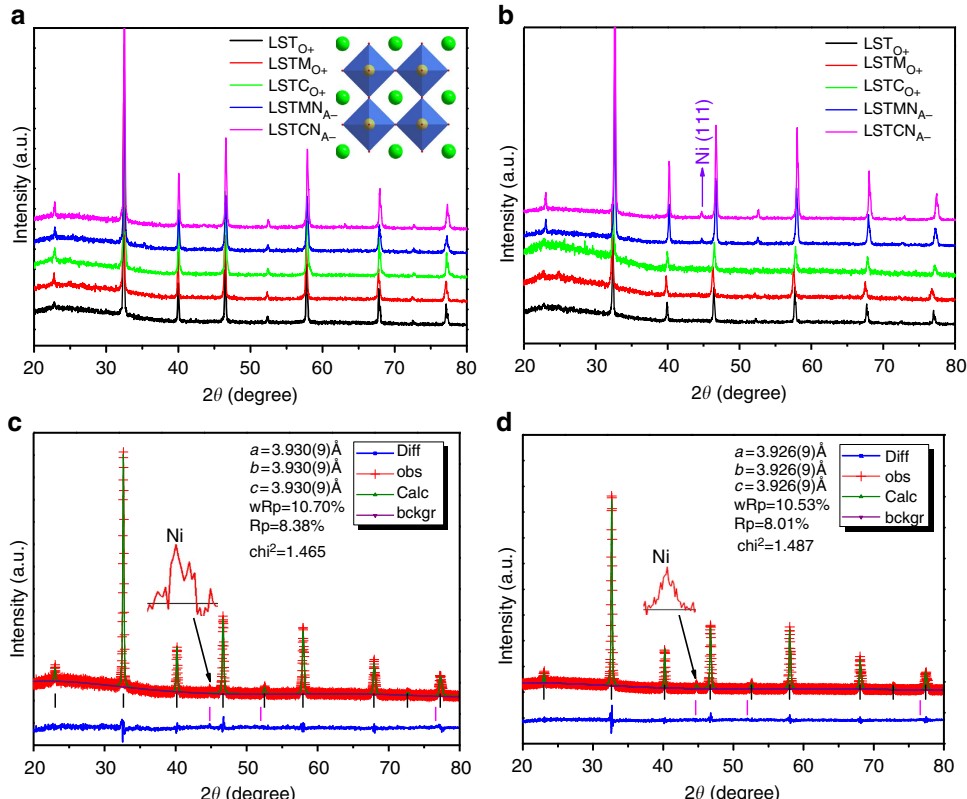

**Figure 1 | X-ray diffraction patterns for a series of samples.** X-ray diffraction of (**a**) oxidized and (**b**) reduced $LST_{O+}$, $LSTM_{O+}$, $LSTC_{O+}$, $LSTMN_{A-}$ and $LSTCN_{A-}$; X-ray diffraction Rietveld refinement of reduced (**c**) $LSTMN_{A-}$ and (**d**) $LSTCN_{A-}$. bckgr, background; calc, calculated; diff, difference; obs, observed.

A clear hetero-junction is visible between the two phases, suggesting a strong interaction between the titanate backbone and exsolved nanoparticles. This anchoring effect is expected to provide enhanced thermal stability of the catalyst, preventing the severe long-term agglomeration which is commonly observed for impregnated catalysts[23]. Regeneration of the nanoparticles is also possible through periodic redox cycling. Besides, selected-area electron diffraction analysis for the reduced $LSTMN_{A-}$ is shown in Supplementary Fig. 5, it shows that $LSTMN_{A-}$ is able to produce the polycrystalline structure.

**Oxygen nonstoichiometry and conductivity.** Thermogravimetric analysis on the various titanates, as shown in Supplementary Fig. 6 and Supplementary Table 2, suggests varying degrees of oxygen stoichiometry, depending on titanium substitution with Cr/Mn/Ni. Whereas $LST_{O+}$ shows a change of 0.032 mol of oxygen upon oxidation/reduction, the Cr- and Mn-substituted stoichiometries release/absorb 0.039 and 0.056 mol of lattice oxygen, confirming the reducibility of the Mn and consequent formation of additional oxygen vacancies upon reduction. However, the $Cr^{3+}$ in titanate lattice is less reducible that leads to similar oxygen absorption for reduced $LST_{O+}$ and $LSTC_{O+}$ samples. And strong reduction only produces partial transition of $Cr^{3+}$ to $Cr^{2+}$ as confirmed by the larger weight gains in thermogravimetric analysis (TGA) in Supplementary Fig. 6c. The weight changes increase even further upon nickel introduction and it is suggested that lattice nickel was partially reduced to metallic nickel during reduction at 800 °C for 20 h. This would result in reduced titanate with Ni present as secondary phase. Assuming that the majority of metallic Ni resides on the titanate

surface, it is clear that an active nanostructure is formed with the presence of Ni nanoparticles coupled with a large concentration of oxygen vacancies.

The varying oxygen contents and abilities to store and release lattice oxygen also manifest themselves in the oxide ionic conductivities of the titanates. The ion conductivity of sintered pellets is tested using the electron-blocking electrode method for the reduced states[24]. As shown in Supplementary Fig. 7, reduced $LST_{O+}$ has the lowest ionic conductivity, reaching $6.8 \times 10^{-4}$ S cm$^{-1}$ at 800 °C. Upon substitution with Cr/Mn and consequently increasing oxygen vacancy concentration under reducing conditions, this increases by an order of magnitude to $6.1–6.3 \times 10^{-3}$ S cm$^{-1}$. Introducing A-site deficiency and nickel on the B-site causes an additional 30% increase to $8.6–9.3 \times 10^{-3}$ S cm$^{-1}$ under identical conditions. These experimental observations are further substantiated by theoretical calculations on oxide ion migration in simplified $SrTiO_3$. These reveal that oxide ion transport by a vacancy-mediated mechanism is much more favourable than through an interstitial one; as shown in Supplementary Figs 7 and 8, the energy barrier for the former is 0.42 eV, as compared to 5 eV for the latter. This is to be expected for the dense perovskite structure and confirms that creating additional oxygen ion vacancies should indeed be beneficial for ionic transport by increasing the concentration of charge carriers.

**$CO_2$ adsorption and activation.** The chemical adsorption of $CO_2$ molecules on reduced titanate samples was investigated by *in situ* infrared spectroscopy. The infrared scans for all the samples without $CO_2$ adsorption at room temperature are shown

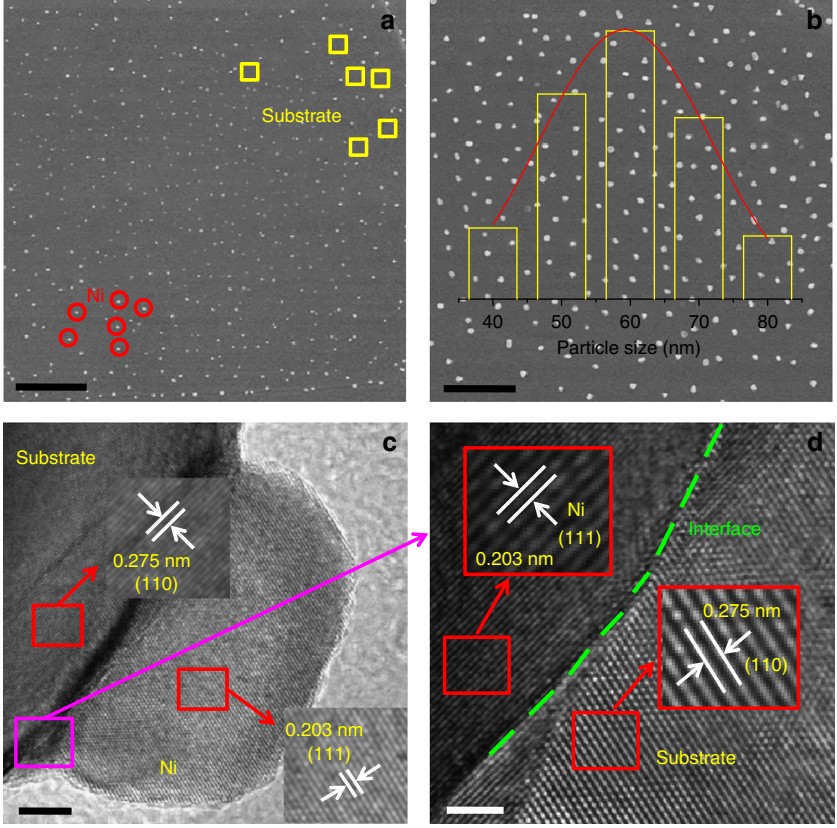

**Figure 2 | Scanning electron microscopy and high-resolution transmission electron microscopy for LSTMN$_{A-}$ sample. (a)** Scanning electron microscopic results for the reduced LSTMN$_{A-}$, Scale bars, 5 μm; **(b)** Ni nanoparticles size distribution of the reduced LSTMN$_{A-}$ inset, Scale bars, 3 μm; **(c)** High-resolution transmission electron microscopic results for the reduced LSTMN$_{A-}$, Scale bars, 5 nm; **(d)** the legible hetero-junction of Ni and substrate along the corresponding to pink square shown in (**c**), Scale bars, 2 nm.

in Supplementary Fig. 9a,b, in which no $CO_2$ or carbonate species ($CO_3^{2-}$) signals are observed. In contrast, all samples with adsorbed $CO_2$ show absorption signals in two distinct infrared bands at room temperature in Supplementary Fig. 9c,d, that is, 2,380–2,300 cm$^{-1}$, which is associated with $CO_2$ molecules[25] on the sample surface, and 1,500–1,430 cm$^{-1}$, which is typically observed for $CO_3^{2-}$ (ref. 26). The chemisorbed carbon dioxide on the titanate surface is therefore assumed to be intermediate between molecular $CO_2$ and carbonate ions. As expected from the concentration of oxygen ion vacancies, which should translate into an active oxide surface, the strongest infrared absorption is observed for LSTMN$_{A-}$ and LSTCN$_{A-}$. To test the ability to adsorb $CO_2$ at elevated temperature, LSTMN$_{A-}$ was subjected to *in situ* infrared tests between 400 and 1,200 °C. As Fig. 3a,b reveal, both infrared bands remain visible up to 1,200 °C, suggesting that both $CO_2$ and carbonate species remain adsorbed, due to strong bonding with this titanate's surface. Temperature Programmed Desorption sheds further light on the adsorption/desorption behaviour. Figure 3c shows how physisorbed $CO_2$ is desorbed below 100 °C, whereas chemisorbed $CO_2$ is retained until much higher temperatures. For Cr/Mn-substituted titanate, the strongest desorption is extended to approximately 800 °C, with concomitant increased adsorption capacity by an order of magnitude as compared to LST$_{O+}$ (0.043 versus 0.0056 ml m$^{-2}$, respectively). These results are further corroborated by TGA in pure Ar atmosphere on reduced titanate samples that have subsequently been treated in $CO_2$ at room temperature for 60 min, shown in Supplementary Fig. 9e. Much enhanced weight loss and continued desorption between 800 and 1,200 °C is observed for all Cr/Mn-substituted samples as

compared to LST$_{O+}$, which desorbs almost all $CO_2$ below 800 °C. This is obviously also much improved when compared to amine-grafted oxides, which typically desorb below 500 °C (refs 20,21).

Theoretical calculations were carried out to construct a tentative mechanism for $CO_2$ adsorption and activation on the titanate surface in conjunction with exsolved Ni nanoparticles and oxide vacancy defects. Before considering the presence and effect of such oxide vacancies however, we first study the $CO_2$ chemisorption behaviour on the boundary of a Ni cluster and $SrTiO_3$ pristine surface (110), see Supplementary Fig. 10. As shown in Fig. 4a and Supplementary Fig. 11c, upon adsorption, low energy scenarios are found for $CO_2$ forming a bidentate configuration including a carbonate formation (Supplementary Fig. 11a). Lowest adsorption energies of $-2.23$ and $-2.60$ eV are calculated for the configurations in which the carbon atom of $CO_2$ binds with a Ni atom from the cluster, whereas one of $CO_2$'s oxygen atoms (O1) attaches to an *hcp* site on the titanate surface, interacting with two Sr atoms and one Ti atom. With a calculated bond length of 1.906 Å for C–Ni, a strong interaction is expected. The Sr–O1 and Ti–O1 distances are 2.490 Å and 2.194 Å, respectively, which differ substantially from bulk strontium titanate (2.767 Å for Sr–O (ref. 27) and 1.952 Å for Ti–O from experimental observations[28]). The C–O1 bond length is elongated to 1.403 Å and the O–C–O angle is bent to 120.9° relative to the gaseous values of 1.18 Å and 180.0°, respectively. This suggests that apart from a strong interaction between adsorbed $CO_2$ and the Ni/$SrTiO_3$ surface, the distorted $CO_2$ seems highly activated. Furthermore, by binding another $CO_2$ molecule to the previously adsorbed oxygen atoms of $CO_2$, such as O2 atom in Fig. 4a, these bidentate configurations could

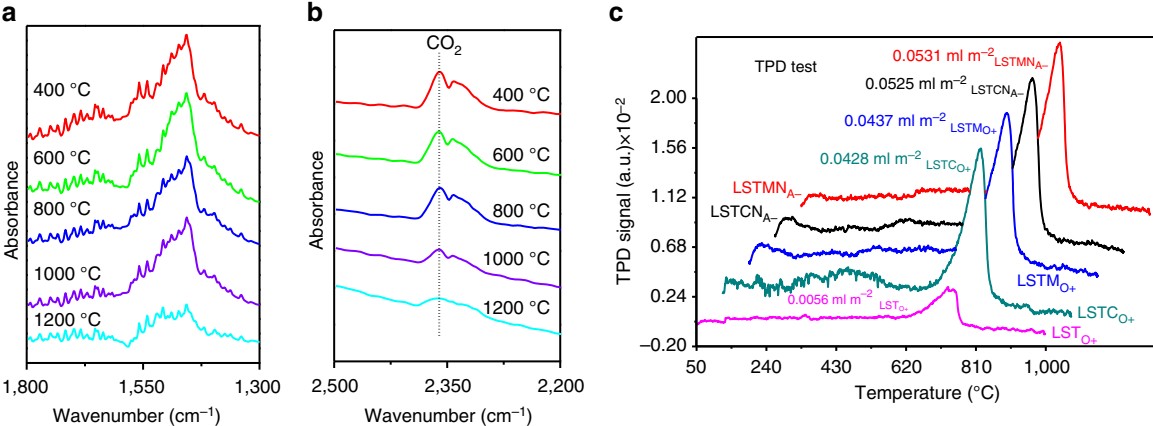

**Figure 3 | Chemical adsorption of $CO_2$ for a series of samples.** (**a,b**) *In situ* infrared spectroscopy of $CO_2$ adsorbed on the reduced $LSTMN_{A-}$ from 400 to 1,200 °C; (**c**) Temperature Programmed Desorption (TPD) test of $CO_2$ on the reduced samples from 50 to 1,000 °C in pure $CO_2$.

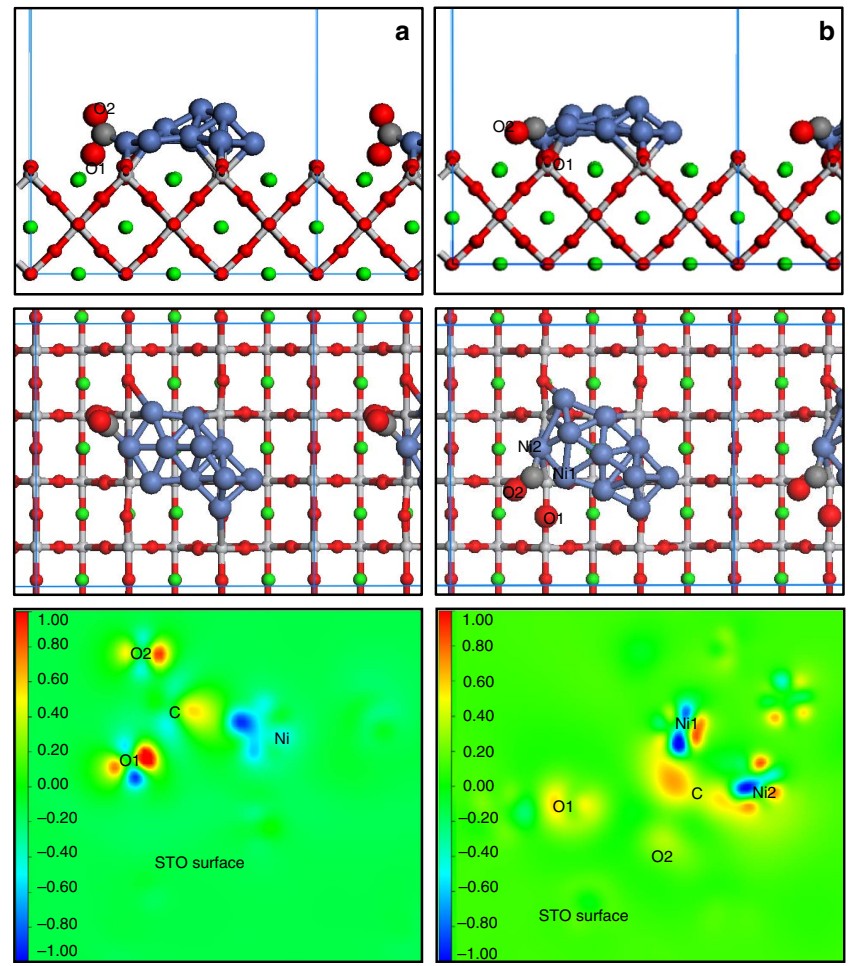

**Figure 4 | Theoretical calculations.** Different adsorption configurations of $CO_2$ on (**a**) the (110) Ni/STO system surface and (**b**) defected site of the (110) Ni/STO system surface. The upper panels show side views while the middle panels give top–down views. Unit cells used in the calculations are marked by blue rectangles. Nickel is blue, strontium is green, titanium is pale, oxygen is red and carbon is grey. The bottom panels show contour plots of the electronic charge density difference for $CO_2$ adsorption on the (110) Ni/STO system surface (bottom left) and defected site of the (110) Ni/STO system surface (bottom right).

be basic configurations to further generate carbonate, which may play a role as shown in our infrared experimental results. Contour plots shown in Fig. 4a indicate that the Ni cluster donates electron

charge to the adsorbed $CO_2$ molecule; the charge density changes take place predominantly within the 2*p* orbitals of C, O, Ni and relevant surface atoms. The electrostatic interactions between

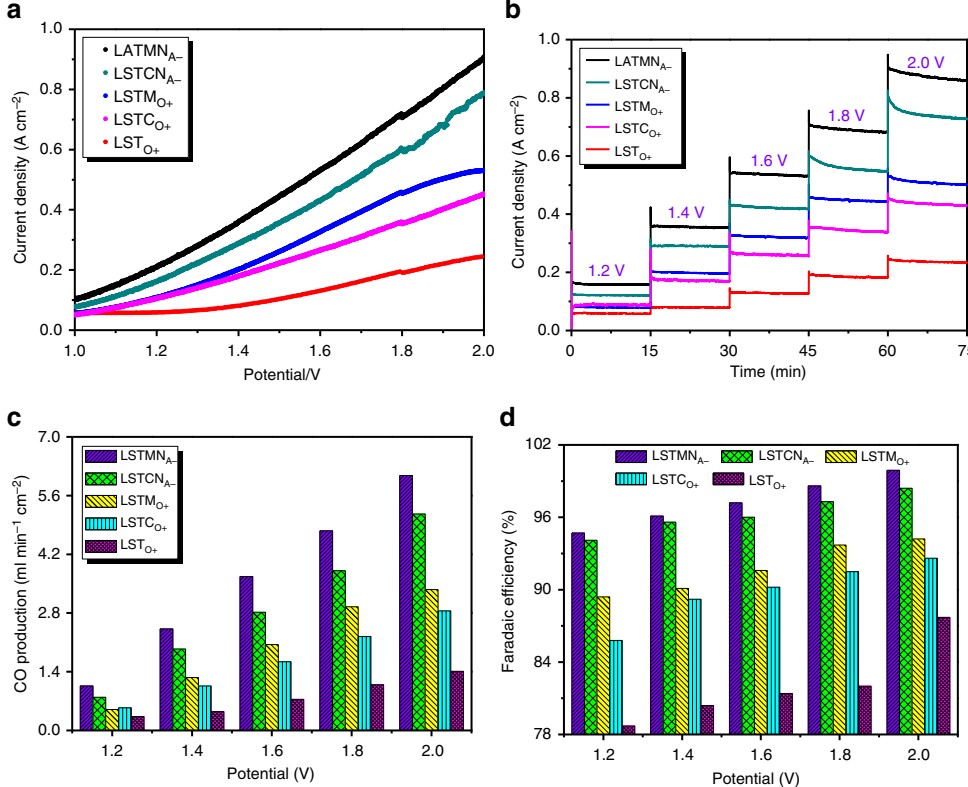

**Figure 5 | Electrolysis performance.** (**a**) *I–V* curves of $CO_2$ electrolysis with different cathodes at 800 °C; (**b**) short-term performances of $CO_2$ electrolysis at different voltages; (**c**) CO production and (**d**) current efficiency with different cathodes.

surface and $CO_2$, which would also affect the observed bidentate configuration, clearly suggest a strong chemical adsorption and activation, promoted by Ni exsolution.

Next we consider adsorption onto a titanate surface that includes oxygen vacancy defects. We use a simplified scenario of a single oxide vacancy site residing in close proximity to the Ni cluster on the $SrTiO_3$ (110) surface and simulate how $CO_2$ would approach and adsorb onto this site as shown in Supplementary Fig. 12. A distinctly different adsorption mechanism emerges now as compared to the vacancy-free titanate surface: in this scenario the $CO_2$ molecule exothermically dissociates into a CO moiety binding to the Ni cluster and an oxygen atom occupying the vacant surface site, with greatly enlarging of C–O1 distance from gaseous value of 1.177 to 3.265 Å (Fig. 4b and Supplementary Fig. 12). The carbon atom of $CO_2$ is now binding to two Ni atoms, with bond lengths C–Ni1 of 1.820 and C–Ni2 of 1.889 Å. The calculated Sr–O1 and Ti–O1 bond distances are 2.728–2.783 Å and 2.020–1.880 Å, respectively, which are very similar to those observed in bulk $SrTiO_3$ (2.767 and 1.952 Å, respectively[27,28], indeed suggesting that O1 is now occupying the oxygen vacancy. The contour plot for this scenario shows that charge density is increased on 2$p$ orbital of C and O1 atom and depleted on Ni 4$s$ orbitals. These charge rearrangements mainly result from the C–O1 bond breaking that causes charge transfer back to the C and O1 atom, respectively. Moreover, electron charge density is also donated by the Ni cluster to the C 2$p$ orbital, albeit originating from Ni 4$s$ orbitals instead. The 2$p$ orbital of O1 accepts charge density from the neighbouring surface atoms. These calculations strongly suggest a synergistic effect from the exsolved Ni cluster and neighbouring oxygen vacancy, causing promotion of the adsorbed $CO_2$ molecule by effectively weakening the C–O1 bond. This

promotion effect is expected to translate itself into a very catalytically active surface for $CO_2$ electrolysis.

**Electrolysis performance.** Both symmetrical and full electrolyser cells were fabricated, comprising composite titanate cathodes with $Ce_{0.8}Sm_{0.2}O_{2-\delta}$. As can be seen in Supplementary Fig. 13, the cells contain dense electrolytes as well as porous electrodes of around 10 μm in thickness. AC impedance was performed on the symmetrical cells to assess the electrode polarization behaviour under reducing conditions, with $pH_2$ ranging from 0.20 to 1.0 bar. For all titanates, it was found that the series resistance $R_s$ was dominated by the YSZ electrolyte, confirming sufficient electronic conductivity as provided by the reduced titanates. And the ohmic resistance of electrode is negligible because the conductivity of electrodes is 3–4 orders of magnitude higher than that of the YSZ electrolyte. The electrode performance of all materials improved significantly with increasing $pH_2$ from 0.20 to 1.0 bar, with polarization resistances decreasing by 50–60%. $LST_{O+}$ was found to have the poorest performance (5.32–2.00 Ω cm²), whereas Cr/Mn-substituted and Ni-exsolving $LSTMN_{A-}$ and $LSTCN_{A-}$ cathodes perform best, with the lowest $R_p$ recorded of 0.67 Ω cm² at 800 °C and $pH_2 = 1.0$ bar. The symmetrical cell results are summarized in Supplementary Figs 14–16.

The cells were used to perform direct $CO_2$ electrolysis under varying applied voltages, ranging from 1.2 to 2.0 V at 800 °C. The open circuit voltage of the cells was established by exposing the cathode and anode to 100% $H_2$ and static air, respectively, and was found to reach 1.1 V, indicating good gas separation by the cells. Typical current–voltage (*I–V*) curves for the various cathode materials in direct $CO_2$ electrolysis mode can be found in Fig. 5. As expected from the previous findings, $LST_{O+}$ shows the poorest

**Table 1 | Performance with different cathodes.**

| Cathode | Applied bias (V) | | | | |
|---|---|---|---|---|---|
| | 1.2 | 1.4 | 1.6 | 1.8 | 2.0 |
| $LST_{O+}$ | | | | | |
| Current density (A cm$^{-2}$) | 0.06 | 0.08 | 0.13 | 0.19 | 0.23 |
| CO production (ml min$^{-1}$ cm$^{-2}$) | 0.33 | 0.45 | 0.74 | 1.09 | 1.41 |
| Faradaic efficiency (%) | 78.7 | 80.4 | 81.4 | 82.0 | 87.7 |
| $LSTM_{O+}$ | | | | | |
| Current density (A cm$^{-2}$) | 0.08 | 0.20 | 0.32 | 0.45 | 0.51 |
| CO production (ml min$^{-1}$ cm$^{-2}$) | 0.50 | 1.26 | 2.05 | 2.95 | 3.36 |
| Faradaic efficiency (%) | 89.4 | 90.1 | 91.6 | 93.7 | 94.2 |
| $LSTC_{O+}$ | | | | | |
| Current density (A cm$^{-2}$) | 0.09 | 0.17 | 0.26 | 0.35 | 0.44 |
| CO production (ml min$^{-1}$ cm$^{-2}$) | 0.54 | 1.06 | 1.64 | 2.24 | 2.85 |
| Faradaic efficiency (%) | 85.8 | 89.2 | 90.2 | 91.5 | 92.6 |
| $LSTM_{A-}$ | | | | | |
| Current density (A cm$^{-2}$) | 0.10 | 0.22 | 0.36 | 0.51 | 0.66 |
| CO production (ml min$^{-1}$ cm$^{-2}$) | 0.63 | 1.39 | 2.34 | 3.41 | 4.43 |
| Faradaic efficiency (%) | 90.1 | 90.4 | 93.0 | 95.6 | 96.0 |
| $LSTC_{A-}$ | | | | | |
| Current density (A cm$^{-2}$) | 0.09 | 0.20 | 0.32 | 0.45 | 0.55 |
| CO production (ml min$^{-1}$ cm$^{-2}$) | 0.56 | 1.26 | 2.06 | 2.98 | 3.67 |
| Faradaic efficiency (%) | 89.0 | 90.1 | 92.1 | 94.7 | 95.4 |
| $LSTMN_{A-}$ | | | | | |
| Current density (A cm$^{-2}$) | 0.16 | 0.36 | 0.54 | 0.69 | 0.87 |
| CO production (ml min$^{-1}$ cm$^{-2}$) | 1.06 | 2.42 | 3.67 | 4.76 | 6.08 |
| Faradaic efficiency (%) | 94.7 | 96.1 | 97.2 | 98.6 | 99.9 |
| $LSTCN_{A-}$ | | | | | |
| Current density (A cm$^{-2}$) | 0.12 | 0.29 | 0.42 | 0.56 | 0.75 |
| CO production (ml min$^{-1}$ cm$^{-2}$) | 0.79 | 1.94 | 2.82 | 3.81 | 5.16 |
| Faradaic efficiency (%) | 94.1 | 95.6 | 96.0 | 97.3 | 98.4 |

$LST_{O+}$, $La_{0.2}Sr_{0.8}Ti_{1.0}O_{3+\delta}$; $LSTC_{A-}$, $(La_{0.2}Sr_{0.8})_{0.95}Ti_{0.9}Cr_{0.1}O_{3-\delta}$; $LSTC_{O+}$, $La_{0.2}Sr_{0.8}Ti_{0.9}Cr_{0.1}O_{3+\delta}$; $LSTCN_{A-}$, $(La_{0.2}Sr_{0.8})_{0.95}Ti_{0.85}Cr_{0.1}Ni_{0.05}O_{3+\delta}$; $LSTM_{A-}$, $(La_{0.2}Sr_{0.8})_{0.95}Ti_{0.9}Mn_{0.1}O_{3-\delta}$; $LSTM_{O+}$, $La_{0.2}Sr_{0.8}Ti_{0.9}Mn_{0.1}O_{3+\delta}$; $LSTMN_{A-}$, $(La_{0.2}Sr_{0.8})_{0.95}Ti_{0.85}Mn_{0.1}Ni_{0.05}O_{3+\delta}$.
Comparison of $CO_2$ electrolysis with different cathodes under applied biases.

performance, reaching a maximum current density of 0.24 A cm$^{-2}$ at 2.0 V. Improved performance is observed for Mn/Cr-substituted compositions, which can be attributed to enhanced $CO_2$ adsorption and activation at high temperatures. $LSTM_{O+}$ reaches a maximum current density of 0.53 A cm$^{-2}$ at 2.0 V. The nickel exsolving compositions show a further improvement, with maximum current densities of 0.91 and 0.79 A cm$^{-2}$ at 2.0 V for $LSTMN_{A-}$ and $LSTCN_{A-}$, respectively. As the potentiostatic measurement in Fig. 5b shows, performances tail off somewhat over time, but a stable current density of 0.87 A cm$^{-2}$ seems to be attained for $LSTMN_{A-}$ at 2.0 V, a factor 3–4 times higher than observed for $LST_{O+}$. Further evidence of these titanates' suitability to reduce $CO_2$ at the cathode comes from inspecting the rate of CO production as measured by gas chromatography and corresponding Faradaic efficiencies at different applied voltages, shown in Fig. 5c,d. $LSTMN_{A-}$ and $LSTCN_{A-}$ yield 6.08 and 5.16 ml min$^{-1}$ cm$^{-2}$ of CO at 2.0 V and 800 °C, respectively, corresponding to Faradaic efficiencies of 99.9 and 98.4%, respectively. When compared to unmodified $LST_{O+}$, this constitutes a 400% improvement of performance. The effect of Ni exsolution yields a performance boost of 75–85% when comparing with Cr/Mn substitution only. Results have been summarized in Table 1.

One aspect that has been ignored so far is the effect of A-site deficiency on electrode performance. To enhance Ni exsolution,

5% A-site deficiency was introduced in the $LSTMN_{A-}$ and $LSTCN_{A-}$ materials, but the A-site deficiency itself may enhance electrode performance. The oxygen interstitial is present in $LST_{O+}$, $LSTM_{O+}$ and $LSTC_{O+}$, though the oxygen vacancies are created and linked to redox active dopants such as Mn/Cr after reduction. However, the A-site deficiency would further produce higher oxygen vacancy concentration that may be positive to electrode activity enhancement. To study this effect, Cr/Mn-substituted $LSTM_{A-}$ and $LSTC_{A-}$ (5% A-site deficiency but no nickel substitution) were also used as cathode materials. Their performance, as indicated in Supplementary Fig. 17, is in between $LSTM_{O+}/LSTC_{O+}$ and $LSTMN_{A-}/LSTCN_{A-}$, suggesting a positive electrocatalytic effect from the A-site deficiency. This may be the result of enhanced reducibility in such compositions and hence oxygen vacancy concentration, as proposed by Savaniu and Irvine[29]. Still, the effect of Ni exsolution adds to the cathode's electrochemical activity, as indicated by the superior performance.

To further understand the observed electrochemical performance in $CO_2$ electrolysis mode at 800 °C for the different cathode materials, impedance spectroscopy was carried out under a range of applied biases, that is, 1.1–1.6 V. Similar trends are observed for all cathode materials, as shown in Supplementary Figs 18 and 19. First, series resistance, $R_s$, is stable and dominated by the electrolyte resistance, which is indicative of the titanates'

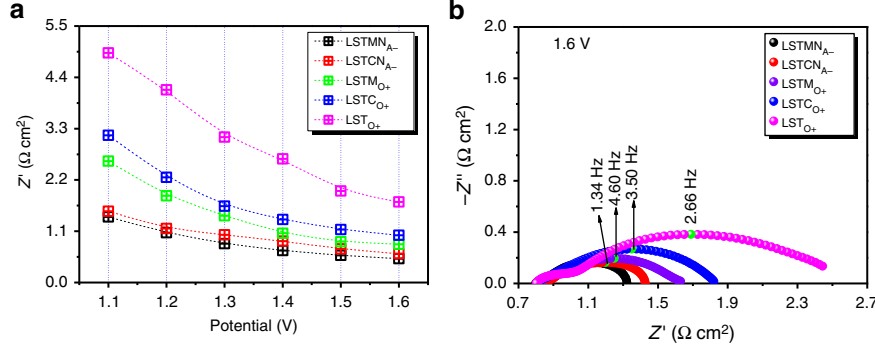

**Figure 6 | AC impedance spectroscopy with different cathodes.** (**a**) The comparison of $R_p$ for the high-temperature $CO_2$ electrolysis with different electrode; (**b**) *in situ* AC impedance at 1.6 V.

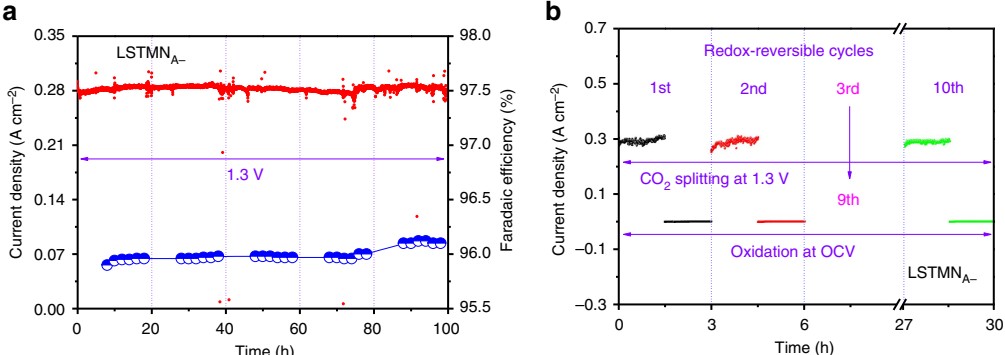

**Figure 7 | Cycling performance.** (**a**) The long-term performance and current efficiency of the $LSTMN_{A-}$ cathode with the flow of $CO_2$ at 800 °C; (**b**) the short-term performance of $LSTMN_{A-}$ cathode after 10 redox cycles.

capability to provide an electronically conductive electrode backbone. In our full electrolysers, the ohmic resistance of electrode is negligible because the conductivity of titanates is 3–4 orders of magnitude higher than that of YSZ electrolyte. Second, polarization resistances, $R_p$, decrease with increasing voltages, which may be due to that the increasing potential leads to a stronger reducing potential that produces higher oxygen vacancy concentration in titanate and thus better ionic conductivity and $CO_2$ adsorption/activation, enhancing electrode kinetics. A change in the fermi level of the electron can also be expected to facilitate electrode polarizations by increasing applied potential. The electrode response seems to be dominated by two low frequency processes. One is related to gas conversion and its response diminishes upon increasing the applied voltage, as expected. The second process is tentatively attributed to dissociative adsorption of reactant species. This seems justified, as $R_p$ is largest for $LST_{O+}$ (4.92–1.73 $\Omega$ cm$^2$ at 1.1–1.6 V) and decreases for $LSTM_{O+}$ (2.86–0.82 $\Omega$ cm$^2$ at 1.1–1.6 V) to finally be lowest for $LSTMN_{A-}$ (1.41–0.51 $\Omega$ cm$^2$ at 1.1–1.6 V). Therefore $R_p$ seems to decrease with the electrode's increasing ability to chemically adsorb and activate $CO_2$, with the best electrode performance observed for electrodes with both exsolved Ni and oxygen vacancies on its surface. The heterojunction interface with Ni nanoparticles interacting with oxygen vacancy on titanate provides a synergy to facilitate electrode reaction. However, the catalytic activity of Mn itself in $ABO_3$ oxide is better than the Cr-doped sample, and the ionic conductivity of $LSTM_{O+}$ is higher than that of $LSTC_{O+}$. The above two points would accordingly deliver better performance for $LSTM_{O+}$ electrode even though $LSTC_{O+}$

has a higher oxygen vacancy concentration. The different cathode materials' impedance responses at 1.6 V are summarized in Fig. 6.

The long-term stability of the exsolving cathode material $LSTMN_{A-}$ was assessed by performing $CO_2$ electrolysis for a period of 100 h at 1.3 V and 800 °C in 100% $CO_2$ atmosphere. As shown in Fig. 7a, a stable current density of 0.28 A cm$^{-2}$ is retained for the duration of the experiment. The stability of this material upon switching between oxidizing and reducing conditions was similarly assessed by redox cycling at 800 °C in alternating 5% $H_2$/Ar and air atmospheres, with subsequent $CO_2$ electrolysis at 1.3 V. As indicated by Fig. 7b and Supplementary Fig. 20, no performance degradation can be observed after 10 such redox cycles, demonstrating excellent stability of the active nanostructured surface of the titanate within the frames of these experiments. The anchoring of Ni nanoparticles to the perovskite lattice is expected to improve both long-term and redox stability, by preventing particle coalescence driven by surface energy reduction[18].

## Discussion

In conclusion, we have shown exceptionally high performance for direct $CO_2$ electrolysis with near 100% Faradaic efficiency. The key is the tailored surface structure of doped strontium titanate that is *in situ* constructed through a combination of Ni particle exsolution and oxygen stoichiometry engineering. These active surface structures enable high-temperature chemical adsorption/activation of $CO_2$ and furthermore exhibit high-temperature stability for several tens of hours with significant

redox cycling ability. Our work shows that these titanates are serious candidates for alternative cathode materials in SOE cells, providing the tools to develop the next generations of electro-chemical devices for energy conversion and storage.

## Methods

**Synthesis.** Ceramic oxides including $La_{0.2}Sr_{0.8}Ti_{1.0}O_{3+\delta}$ ($LST_{O+}$), $La_{0.2}Sr_{0.8}Ti_{0.9}Cr_{0.1}O_{3+\delta}$ ($LSTC_{O+}$), $La_{0.2}Sr_{0.8}Ti_{0.9}Mn_{0.1}O_{3+\delta}$ ($LSTM_{O+}$), $(La_{0.2}Sr_{0.8})_{0.95}Ti_{0.9}Mn_{0.1}O_{3+\delta}$ ($LSTM_{A-}$), $(La_{0.2}Sr_{0.8})_{0.95}Ti_{0.9}Cr_{0.1}O_{3-\delta}$ ($LSTC_{A-}$), $(La_{0.2}Sr_{0.8})_{0.95}Ti_{0.85}Cr_{0.1}Ni_{0.05}O_{3+\delta}$ ($LSTCN_{A-}$), $(La_{0.2}Sr_{0.8})_{0.95}Ti_{0.85}Mn_{0.1}Ni_{0.05}O_{3+\delta}$ ($LSTMN_{A-}$), $(La_{0.8}Sr_{0.2})_{0.95}MnO_{3-\delta}$ ($LSM_{A-}$) and $Ce_{0.8}Sm_{0.2}O_{2-\delta}$ powders were synthesized using a solid-state reaction method performed in air[30].

**Characterization.** Phase formations were confirmed by using X-ray diffraction (Cu K$\alpha$, Miniflex 600, Rigaku Corporation, Japan), and the data were refined by using the General Structure Analysis System software[31]. Scanning electron microscopy (SU-8010, JEOL Ltd, Japan) and high-resolution transmission electron microscopy (Tecnai F20, FEI Ltd, USA) were employed to investigate the exsolution of nanoparticles. XPS (ESCALAB 250Xi, Thermo, USA) with monochromatized Al K$\alpha$ at $h\nu = 1{,}486.6$ eV was utilized to analyse elemental oxidation states. TGA was conducted on a Netzsch STA449F3 to calculate oxygen non-stoichiometry. To this end, powder samples were first reduced in 5% $H_2$/Ar at 800 °C for 20 h and TGA was subsequently performed on these samples from room temperature to 1,200 °C in air. The electrical properties of the reduced samples were examined using DC four-terminal method (Keithley 2,000, Keithley Instruments Inc., USA) in an atmosphere of 5% $H_2$/Ar between 650 and 800 °C. The $H_2O$ content in 5%$H_2$/Ar or $H_2$ streams were measured to be 0.5 and 1%, respectively corresponding to oxygen partial pressures of $\sim 1 \times 10^{-20}$ and $\sim 1 \times 10^{-22}$ atm, respectively, at 800 °C. Approximately 2.0 g of the samples' powders was pressed into pellets and sintered at 1,400 °C for 10 h in air to get dense samples for the conductivity tests. Before the conductivity tests, the samples' pellets were reduced at 800 °C for 20 h in 5% $H_2$/Ar. All infrared spectra were collected using Fourier transform infrared spectrometer (VERTEX 70, Bruker). The powder samples were first reduced in 5% $H_2$/Ar at 800 °C for 20 h and Temperature Programmed Desorption tests of $CO_2$ were subsequently recorded from 50 to 1,200 °C in $CO_2$ with a Micromeritics-Hiden Autochem II 2920-QIC20.

**Electrochemical characterization.** Symmetric cells with 0.5-mm-thick YSZ and different titanate electrodes (1 cm$^2$) were assembled and treated at 1,200 °C for 3 h in air. The current collector was made with silver paste (SS-8060, Xinluyi, China) and treated at 550 °C for 30 min in air. The 0.5-mm-thick YSZ-supported single SOEs with titanate cathodes (1 cm$^2$) and LSM anode (1 cm$^2$) were assembled and treated at 1,200 °C for 3 h in air. The current collector was made with silver paste (SS-8060, Xinluyi, China) and treated at 550 °C for 30 min in air. Electrochemical measurement was performed using an electrochemical station (IM6, Zahner, Germany). The frequency range was 4 MHz to 100 mHz, and the voltage perturbation was 10 mV. The gas flow was controlled with mass flow meters (D08-3 F, Sevenstar, China). The electrolysis experiment was conducted by flowing 5% $H_2$/Ar at the flow rate of 50 ml min$^{-1}$ for 1.5 h and then pure hydrogen at the flow rate of 50 ml min$^{-1}$ was supplied to cathode for 2.0 h at operation temperature. After that, the cathode would be sufficiently reduced and activated. And then the $CO_2$ electrolysis was performed. The online gas chromatography (GC2014, Shimazu, Japan) was used to analyse the CO production of the output gas from the electrolyser cells at the flow rate of 50 ml min$^{-1}$ at 800 °C.

**Theoretical calculations.** Theoretical calculations are utilized to understand the mechanism of oxide ion conduction and $CO_2$ chemical adsorption. All calculations were performed using density functional theory implemented in the Vienna Ab Initio Simulation Package (VASP)[32,33]. Within the projector augmented wave (PAW) framework, the plane-wave cutoff energy was set to 500 eV, which gives well-converged relative energies for the system. The Perdew–Burke–Ernzerhof functional was used to describe exchange and correlation[34]. In electronic and geometric optimizations, energies and residual forces were converged to $10^{-6}$ eV and 0.02 eV Å$^{-1}$, respectively. In order to simplify and facilitate the calculation, the creation of oxygen vacancies through La and Mn/Cr substitution on the A and B-site, respectively, has been simplified by making oxygen-deficient $SrTiO_3$. The lattice parameters of $SrTiO_3$ (STO), optimized with a $13 \times 13 \times 13$ k-point grid[35], were $a = b = c = 3.947$ Å, which is in good agreement with our experimental values. A superstructure with dimensions of $11.86 \times 11.86 \times 7.89$ Å$^3$ (89 atoms) was used to simulate the oxygen transport process within STO bulk. A $2 \times 2 \times 4$ k-point grid was used for Brillouin zone sampling of this superstructure model. The climbing-image nudged elastic band method[36] was used to explore the transition states of the oxygen vacancies transport process. The periodic slab model was used to simulate the (110) surface of STO with Sr-Ti-O terminations. For this model, three Sr-Ti-O layers and two O layers were included in the slab. The three bottom layers were fixed to their bulk geometries during optimization while other atoms were fully relaxed. A vacuum layer with thickness of about 22 Å was inserted in the $c$ direction to avoid the slab-to-slab interaction. The Ni segregation on the (110) surface slab of STO was mimicked by a system containing a Ni cluster with 11 Ni atoms laying on (110) surface of STO with Sr-Ti-O terminations. A $2 \times 2 \times 1$ k-point grid is employed for Brillouin zone sampling of (110) Ni/STO surface system. The adsorption energy of $CO_2$ is calculated as $E = E_{tot} - E_{slab} - E_{ad}$, where $E_{tot}$, $E_{slab}$ and $E_{ad}$ are the energy of the Ni/STO surface system without adsorption, the energy of the $CO_2$ in gas phase and the total energy of the adsorption system, respectively.

**Data availability.** The data that support the findings of this study are available from the corresponding author(s) upon request.

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

## Acknowledgements

K.X. acknowledges Natural Science Foundation of China (91545123) and Natural Science Foundation of Fujian Province (2016J01275) for funding this work. C.L. acknowledges support by the Strategic Priority Research Program of the Chinese Academy of Sciences, Grant No. XDB20000000 and Hundred Talents Program of the Chinese Academy of Sciences. J.T.S.I. acknowledges funding from EPSRC Platform Grant EP/K015540/1 and Royal Society Wolfson Merit Award WRMA 2012/R2.

## Author contributions

K.X. fully designed and supervised the project. J.T.S.I. supervised part of the project. L.Y. conducted the experimental work. M.Z. and C.L. carried out the theoretical calculations. All authors contributed to data analysis and gave approval to the final version of the manuscript.

## Additional information

**Competing interests:** The authors declare no competing financial interests.

**Publisher's note**: 

