## [Peer Review File · Nature Communications]

Reviewers' comments:

Reviewer #1 (Remarks to the Author):

The authors present an interesting cathode for CO₂ electrolysis in a Solid Oxide Electrolyser Cell, especially the exsolution of catalytically active Ni particles under reducing conditions is remarkably effective and apparently reversible. This is an important finding that will fit in the scope of Nature Communications.

There is, however, a serious problem with the manuscript that needs to be repaired before it can be

published. The authors present the reducing ambient' s as hydrogen in argon, or even pure hydrogen. It

is well known that a proper thermodynamic equilibrium can only be reached when also the partial pressure of H₂O is controlled, either by using something like a water bubbler, or that the background

water pressure is monitored. There is no information on this in the manuscript. Without control of the

partial pressure of water the level of reduction is controlled by the rate of the reduction and the gas

flows.

There are further a few minor points:

- Page 6, line 238: The decrease in the polarisation resistance with increasing potential is a normal effect and is controlled by several factors, among which a change in the fermi level of the electron (-holes). That an increase in oxygen vacancy concentration is the cause is rather unlikely.
- Page 7, line 260, should read: '... with **near** 100% Faradaic efficiency'.
- I question the high degree of accuracy presented in table 1 (page 10). Please use a realistic number of 'significant digits'.
- Page 13, Fig.3: '.. CO₂-TPD analysis of the reduced samples from 50 to 1000°C in **pure CO₂**.'??
- Suppl. Info., Figure S1: figure (a) does not show a difference graph.
- Ibid., Fig. S5 (a) does not show a particle size distribution.
- Ibid., Fig. S7: It is normal to present ionic conductivities in an Arrhenius style graph.
- Ibid., Figure S21 does not add significant information, except for the photograph of the electrolysis cell, but this picture is of poor quality

Reviewer #2 (Remarks to the Author):

This is a good paper. It should be published after minor, but necessary, revisions are made. Most importantly, the authors seem to dismiss the significant ohmic resistance in the electrodes using their materials systems, even at a high operating temperature like 800°C. This is a significant challenge that needs to be addressed for this to be viable. At minimum, the ohmic resistance should be clearly explained. I also recommend the authors describe/propose how they might achieve good ohmic resistance in their electrodes, and if/how that may impact the performance they report here.

There are numerous syntax errors throughout the manuscript that need to be fixed. I mentioned a few below, but there are more.

Other comments:

Line 22. "electrolyser" should be plural.

Line 24. Where does the high temperature heat stream come from?

Line 41. What is meant by "well adapted"?

Line 44. Seems like there should be references should be provided for this statement.

Line 48. The authors need to clarify what is meant by this sentence. The host lattice is the ceramic

and is not providing the catalytic activity. As written, it seems to suggest the host lattice is the catalyst.

Line 51. Remove "an"

Line 51. Remove "In this case,"

Line 100. It would be helpful for the "clear hetero junction" in Figure S5b to be pointed out within the figure. Is this the figure the authors were referring to?

IR Figures: Are the y-axes correct? The plots look like absorbance plots, not transmittance plots.

IR Figures: An IR scan without adsorbed CO₂ would be helpful for reference. As presented, there is no reference point for the data.

Line 174: replace "effect" with "affect"

Lines 235-236. It seems like the authors are suggesting the ohmic resistances of the porous titanate electrodes are essentially zero. Given the conductivities they report, the ohmic resistance of the porous electrodes is quite substantial. Even at 800°C. Even if the electrode thickness was only 10 microns, this corresponds to an ohmic resistance of 0.17 Ω·cm⁻² for each electrode, a total of 0.34 Ω·cm⁻² at 800°C. This is quite substantial at such a high operating temperature. This needs to be explained. Also, I recommend the authors describe/propose how they might achieve good ohmic resistance in the electrodes for their systems, and if/how that may impact their Ni nanoparticles.

Reviewer #3 (Remarks to the Author):

Perovskite structured strontium-doped lanthanum titanates with A site deficiency, B site doped, and Ni nanoparticle exsolved are carefully investigated as the cathodes of electrolyzer for CO₂ splitting. While the work is significant in providing new knowledge to the area of applying CO₂ electrolyzing for energy storage, several issues should be addressed before publication.

1. The authors attribute the improved adsorption of CO₂ on the titanates, with Mn and Cr doping, to the increased oxygen vacancy concentration. However, it seems conflict with the experimental results which obviously show that the oxygen vacancy concentration of the Cr-doped material is larger than that of the Mn-doped one (Table S2, Line 112-116) while the performances of the latter are better than the former (Table 1, Figure 5, and Figure 6, ect.).

Besides, the reduced titanate stoichiometry (La_{0.2}Sr_{0.8})_{0.95}Ti_{0.85}Mn_{0.1}O_{2.858} (Line 112-116) can be rewritten as (La_{0.2}Sr_{0.8}Ti_{0.85}/0.95Mn_{0.1}/0.95)_{0.95}O_{2.85+0.008} or (La_{0.2}Sr_{0.8}Ti_{0.85}/0.95Mn_{0.1}/0.95O₃)_{0.95}O_{0.008}. It means that the A sites, B sites, and the oxygen sites of the perovskite structure are all occupied with excess oxygens. For this, it is not proper to conclude that "an electrochemically active surface is formed with the presence of nanoparticles in conjunction with a large concentration of oxygen vacancies."

Because of the above two points, the explanations on the functions of oxygen vacancy are not convincing and seem somewhat suspicious.

2. Line 105-111: Figure S6 shows that the weight varies with temperature. What is the temperature used to obtain the data of oxygen change?

3. Figure 4: I am just wondering if the upper panels of the figure should be called as side view

(viewing from the side) while the middle ones top view (viewing from the top)?

4. There are some mistakes in the manuscript, as pointed out in the following:

Line 79: A comma should follow (LSTMA-) and (LSTCA-), respectively.

Line 169: "...such as O₂ atom..." The subscript is not correctly used in the manuscript.

Line 174: ...which would also affect...

Line 191-194: Please check the sentences: There is a repetition.

Figure 1a: The pattern of LSTo+ is missing.

Response to reviewers

Reviewer #1

The authors present an interesting cathode for CO₂ electrolysis in a Solid Oxide Electrolyser Cell, especially the exsolution of catalytically active Ni particles under reducing conditions is remarkably effective and apparently reversible. This is an important finding that will fit in the scope of Nature Communications.

Answer: Many thanks for your comments. We have carefully revised this manuscript according to your suggestions.

1. There is, however, a serious problem with the manuscript that needs to be repaired before it can be published. The authors present the reducing ambient's as hydrogen in argon, or even pure hydrogen. It is well known that a proper thermodynamic equilibrium can only be reached when also the partial pressure of H₂O is controlled, either by using something like a water bubbler, or that the background water pressure is monitored. There is no information on this in the manuscript. Without control of the partial pressure of water the level of reduction is controlled by the rate of the reduction and the gas flows.

Answer: Thank you very much and we have added the oxygen partial pressure data in revision. The H₂O content in 5% H₂/Ar or H₂ streams were measured to be 0.5% and 1% respectively corresponding to oxygen partial pressures of $\sim 1 \times 10^{-20}$ and $\sim 1 \times 10^{-22}$ atm, respectively, at 800°C.

2. There are further a few minor points:

- Page 6, line 238: The decrease in the polarisation resistance with increasing potential is a normal effect and is controlled by several factors, among which a change in the fermi level of the electron (-holes). That an increase in oxygen vacancy concentration is the cause is rather unlikely.

Answer: Thank you very much and we have revised the explanation in revision. The increasing potential leads to a stronger reducing potential that produces higher oxygen vacancy concentration in titanate cathode, which may be beneficial to catalytic activity of cathode because of its enhanced ionic transport and CO₂ adsorption/activation. As you said, a change in the fermi level of the electron (holes) can also be expected to enhance electrode performance by increasing applied potential. We have made a proper expression in revision.

- Page 7, line 260, should read: '... with near 100% Faradaic efficiency'.

Answer: Thanks. We have corrected it in revision.

- I question the high degree of accuracy presented in table 1 (page 10). Please use a realistic number of 'significant digits'.

Answer: Many thanks. We corrected the data accuracy in revision.

- Page 13, Fig.3: ‘.. CO₂-TPD analysis of the reduced samples from 50 to 1000°C in pure CO₂.’??

Answer: Thanks. We revised the expression in revision. The powder samples were firstly reduced in 5%H₂/Ar at 800°C for 20 h and Temperature Programmed Desorption (TPD) tests of CO₂ were subsequently performed from 50 to 1200°C in pure CO₂.

- Suppl. Info., Figure S1: figure (a) does not show a difference graph.

Answer: Many thanks. We have corrected in revision.

- Ibid., Fig. S5 (a) does not show a particle size distribution.

Answer: Many thanks. We have carefully corrected it in revision.

- Ibid., Fig. S7: It is normal to present ionic conductivities in an Arrhenius style graph.

Answer: Many thanks. We have carefully revised it in revision.

- Ibid., Figure S21 does not add significant information, except for the photograph of the electrolysis cell, but this picture is of poor quality.

Answer: Thanks. We have deleted it in revision.

Reviewer #2

1. This is a good paper. It should be published after minor, but necessary, revisions are made. Most importantly, the authors seem to dismiss the significant ohmic resistance in the electrodes using their materials systems, even at a high operating temperature like 800°C. This is a significant challenge that needs to be addressed for this to be viable. At minimum, the ohmic resistance should be clearly explained. I also recommend the authors describe/propose how they might achieve good ohmic resistance in their electrodes, and if/how that may impact the performance they report here.

Answer: Many thanks for your comments. The conductivity of electrode materials is 3~4 orders of magnitude higher than YSZ electrolyte, which makes the ohmic resistance of electrode negligible in contrast to electrolyte. The series resistance (R_s) in symmetrical cells, which mainly comes from ionic resistance of the YSZ electrolyte, is generally stable in a wide range of pH_2 . In this case, all the series R_s have been set as 0 to compare the electrode polarization resistances (R_p) in Figure S14-S16. We have added the R_s in Figure S16 in (b) inset according to your suggestion. The ohmic resistances of electrode is still negligible in our full electrolysers.

2. There are numerous syntax errors throughout the manuscript that need to be fixed. I

mentioned a few below, but there are more.

Answer: Many thanks. We have carefully corrected the syntax errors in revision.

Other comments:

Line 22. “electrolyser” should be plural.

Answer: Many thanks. We have carefully corrected it in revision.

Line 24. Where does the high temperature heat stream come from?

Answer: Thanks. We added the heat source in revision. The high-temperature heat stream come from many kinds of fields. The renewable energy resources, such as wind, solar and geothermal energy, can be converted into electricity or heat energy for SOEs. Other alternatives, such as nuclear energy and unused heat from industry, are also important as energy source to conduct electrolysis processes using SOEs at high temperatures. Therefore, renewable electricity and heat source for SOEs technology are sufficient and sustainable without net CO₂ emission and fossil fuel consumption.

Line 41. What is meant by “well adapted”?

Answer: Thanks. We have added explanation in revision. The p-type conduction of oxide material like La_xSr_{1-x}Cr_yMn_{1-y}O_{3-δ} (LSCM) is not ideally adapted to a reducing potential that leads to the decrease in conductivity, chemical and structural changes. The n-type conductivity of reduced titanate like the conductivity of La_{0.2}Sr_{0.8}TiO_{3.1} has a strong dependence on the decreasing oxygen partial pressure which is expected to produce reducing conditions. The La_{0.2}Sr_{0.8}TiO_{3.1} is partially electrochemically reduced (Ti⁴⁺→Ti³⁺) at potentials required for CO₂ reduction and the n-type electronic conduction is accordingly enhanced, which delivers improved cathode performances with favorable kinetics.

Line 44. Seems like there should be references should be provided for this statement.

Answer: Many thanks. We have added the references in revision.

Line 48. The authors need to clarify what is meant by this sentence. The host lattice is the ceramic and is not providing the catalytic activity. As written, it seems to suggest the host lattice is the catalyst.

Answer: Many thanks. We made it different in revision. The perovskite oxides (ABO₃) are employed as supports and thus certain metal element can be incorporated as cations on the B site of the perovskite lattice under oxidizing conditions and partly exsolved as nanoparticles on subsequent reduction, which thus opens the possibility of *in situ* growth of catalysts. Therefore, in our work, active nanostructures are investigated on titanate surface under differing regimes of perovskite non-stoichiometry. The exsolved metal nanoparticles coupled with tailored oxygen

vacancies through Cr and Mn substitution produce a strongly interactive interface.

Line 51. Remove “an” and “In this case,”

Answer: Many thanks. We have removed them in revision.

Line 100. It would be helpful for the “clear hetero junction” in Figure S5b to be pointed out within the figure. Is this the figure the authors were referring to?

Answer: Thanks. The “clear hetero junction” corresponds to Figure 2d. We have clearly pointed it in revision.

IR Figures: Are the y-axes correct? The plots look like absorbance plots, not transmittance plots.

Answer: Many thanks. We apologize for the mistakes and we have carefully corrected them in revision.

IR Figures: An IR scan without adsorbed CO₂ would be helpful for reference. As presented, there is no reference point for the data.

Answer: Many thanks. We have added the IR data and references in revision. The IR scan without adsorbed CO₂ for all samples have been added in Fig.S9 (d) and (e).

Line 174: replace “effect” with “affect”

Answer: Many thanks. We have replaced it in revision.

Lines 235-236. It seems like the authors are suggesting the ohmic resistances of the porous titanate electrodes are essentially zero. Given the conductivities they report, the ohmic resistance of the porous electrodes is quite substantial. Even at 800°C. Even if the electrode thickness was only 10 microns, this corresponds to an ohmic resistance of 0.17 Ω•cm² for each electrode, a total of 0.34 Ω•cm² at 800°C. This is quite substantial at such a high operating temperature. This needs to be explained. Also, I recommend the authors describe/propose how they might achieve good ohmic resistance in the electrodes for their systems, and if/how that may impact their Ni nanoparticles.

Answer: Many thanks for your comments. The series resistances (R_s) in electrolyser cells, including ohmic resistance of electrolyte and electrodes, are included in the ohmic resistance (R_s) as tested using AC impedance method. The R_s is 0.80 Ω•cm² at 1.6 V, which is related to the total ohmic resistance of the cell but mainly comes from the ionic transport in electrolyte. The ohmic resistance of electrode materials is negligible because it is 3~4 orders of magnitude lower than YSZ electrolyte. The R_p , electrode polarization resistance, is a reflection of electrode activity under applied potentials as tested using AC impedance method.

However, the exsolved Ni nanoparticles in electrode are mainly benefit for the electrocatalytic activity. The hetero-junction interface with Ni nanoparticles interacting with defected titanate provides a synergy to facilitate electrode reaction. The catalytic mechanism of CO₂ splitting has been provided in revision.

Reviewer #3

Perovskite structured strontium-doped lanthanum titanates with A site deficiency, B site doped, and Ni nanoparticle exsolved are carefully investigated as the cathodes of electrolyzer for CO₂ splitting. While the work is significant in providing new knowledge to the area of applying CO₂ electrolyzing for energy storage, several issues should be addressed before publication.

Answer: Many thanks for your comments. We have carefully revised this manuscript according to your suggestions.

1. The authors attribute the improved adsorption of CO₂ on the titanates, with Mn and Cr doping, to the increased oxygen vacancy concentration. However, it seems conflict with the experimental results which obviously show that the oxygen vacancy concentration of the Cr-doped material is larger than that of the Mn-doped one (Table S2, Line 112-116) while the performances of the latter are better than the former (Table 1, Figure 5, and Figure 6, ect.).

Answer: Thanks for your comments. We added the explanations in revision. The difference of non-stoichiometry is mainly related to the different oxidation states of Mn or Cr in lattice. However, the catalytic activity of Mn itself in ABO₃ oxide is better than the Cr-doped sample, which is a scientific law widely used to design new electrode materials. In addition, the ionic conductivity of LSTM_{O+} is higher than that of LSTC_{O+}. The above two points would accordingly deliver better performance for LSTM_{O+} electrode even though LSTC_{O+} has a higher oxygen vacancy concentration.

Besides, the reduced titanate stoichiometry (La_{0.2}Sr_{0.8})_{0.95}Ti_{0.85}Mn_{0.1}O_{2.858} (Line 112-116) can be rewritten as (La_{0.2}Sr_{0.8}Ti_{0.85/0.95}Mn_{0.1/0.95})_{0.95}O_{2.85+0.008} or (La_{0.2}Sr_{0.8}Ti_{0.85/0.95}Mn_{0.1/0.95}O₃)_{0.95}O_{0.008}. It means that the A sites, B sites, and the oxygen sites of the perovskite structure are all occupied with excess oxygens. For this, it is not proper to conclude that “an electrochemically active surface is formed with the presence of nanoparticles in conjunction with a large concentration of oxygen vacancies.”

Answer: Many thanks. The reviewer is correct to raise this point, in our original submission we reported samples reduced to 1200°C as these give better TGA traces and more accurate analyses. Unfortunately this was misleading as 800°C is much more relevant to the electrochemistry. We have revised this and the changes in defect chemistry are much less significant so that the overall nature of oxygen excess and oxygen deficient is not perturbed. Of key importance is that much of the Ni remains in the lattice, as has been previously observed.

Because of the above two points, the explanations on the functions of oxygen vacancy are not convincing and seem somewhat suspicious.

Answer: Many thanks. In our work, we present a double doping strategy in titanate cathode to facilitate CO₂ reduction, promoting adsorption/activation by making use of redox active dopants such as Mn linked to oxygen vacancies and dopants such as Ni that afford metal nanoparticle exsolution. The adsorbed and activated CO₂ adopts an intermediate chemical state between a carbon dioxide molecule and a carbonate ion at Ni/titanate interface. The synergistic control of nonstoichiometry and Ni exsolution provides optimal performance for CO₂ electrolysis with no degradation being observed after 100 hours of high temperature operation and 10 redox cycles.

Creation of oxygen vacancy is widely utilized to chemically accommodate and activate CO₂ molecules especially at intermediate to high temperatures. The doping of redox-active Mn/Cr in LSTMO₊/LSTCO₊ creates oxygen vacancies linked to Mn/Cr even though some oxygen interstitials might be still present. The Mn/Cr-doped LSTMO₊/LSTCO₊ cathodes demonstrate enhanced performance in contrast to bare titanate cathode. In addition, the 5% A-site deficient LSTM_{A-}/LSTC_{A-} with higher oxygen vacancy concentrations further demonstrate a better performance.

2. Line 105-111: Figure S6 shows that the weight varies with temperature. What is the temperature used to obtain the data of oxygen change?

Answer: Many thanks. We use the weight gain (%) to determine the oxygen change. The reduced sample is heated in TG analyzer in air from room temperature to 1200°C.

3. Figure 4: I am just wondering if the upper panels of the figure should be called as side view (viewing from the side) while the middle ones top view (viewing from the top)?

Answer: Many thanks. We have corrected it in revision.

4. There are some mistakes in the manuscript, as pointed out in the following:

Line 79: A comma should follow (LSTM_{A-}) and (LSTC_{A-}), respectively.

Line 169: "...such as O₂ atom..." The subscript is not correctly used in the manuscript.

Line 174: ...which would also affect...

Line 191-194: Please check the sentences: There is a repetition.

Figure 1a: The pattern of LST_{o+} is missing.

Answer: Many thanks. We have carefully corrected it in revision.

REVIEWERS' COMMENTS:

Reviewer #1 (Remarks to the Author):

The authors present an interesting cathode concept for CO₂ electrolysis in a Solid Oxide Electrolyser Cell. The cathode is based on (La,Sr)TiO₂ with different dopant additions. The (La,Sr)(Mn,Ni,Ti)O(2- δ) shows the best performance. The exsolution of catalitically active Ni nanoparticles under reducing conditions is remarkably effective. The reversibility of the exsolution in a redox cycle improves the stability of the Ni particles, thus preventing agglomeration in long term use.

The different cathode materials are well characterized. A DFT analysis of CO₂ absorption on small Ni clusters shows that the CO₂ adopts an intermediate state between CO₂ and carbonate ion.

Reviewer #2 (Remarks to the Author):

My comments in the initial review have been sufficiently addressed for publication.

Reviewer #3 (Remarks to the Author):

The issues I raised in the first review process have been well addressed. I recommend publication.

Response to reviewers

Reviewer #1 (Remarks to the Author):

The authors present an interesting cathode concept for CO₂ electrolysis in a Solid Oxide Electrolyser Cell. The cathode is based on (La,Sr)TiO₃ with different dopant additions. The (La,Sr)(Mn,Ni,Ti)O_{3-δ} shows the best performance. The exsolution of catalytically active Ni nano-particles under reducing conditions is remarkably effective. The reversibility of the exsolution in a redox cycle improves the stability of the Ni particles, thus preventing agglomeration in long term use. The different cathode materials are well characterized. A DFT analysis of CO₂ absorption on small Ni clusters shows that the CO₂ adopts an intermediate state between CO₂ and carbonate ion.

Answer: Thank you very much for your comment.

Reviewer #2 (Remarks to the Author):

My comments in the initial review have been sufficiently addressed for publication.

Answer: Thanks very much for your comment and suggestion.

Reviewer #3 (Remarks to the Author):

The issues I raised in the first review process have been well addressed. I recommend publication.

Answer: Thanks for your comment and suggestion.